# Electron-Enriched Pd Nanoparticles for Selective Hydrogenation of Halonitrobenzenes to Haloanilines

Zechen Liang [1], Mingkai Zhang [1], Sai Zhang [2],* and Yongquan Qu [1,2,*]

[1] Frontier Institute of Science and Technology, Xi'an Jiaotong University, Xi'an 710049, China; xj3118128001@stu.xjtu.edu.cn (Z.L.); zmk0102@stu.xjtu.edu.cn (M.Z.)

[2] Key Laboratory of Special Functional and Smart Polymer Materials of Ministry of Industry and Information Technology, School of Chemistry and Chemical Engineering, Northwestern Polytechnical University, Xi'an 710072, China

* Correspondence: zhangsai1112@nwpu.edu.cn (S.Z.); yongquan@mail.xjtu.edu.cn (Y.Q.)

**Abstract:** Selective hydrogenation of halonitrobenzenes into haloanilines represents a green process to replace the environmentally unfriendly non-catalytic chemical reduction process in industry. However, this transformation often suffers from serious dehalogenation due to the easy break of carbon-halogen bonds on metal surfaces. Modulations of the electronic structure of the supported Pd nanoparticles on Lewis-basic layered double hydroxides have been demonstrated to promote catalytic activity and selectivity for hydrogenation of halonitrobenzenes into haloanilines. Mechanism studies suggest that Pd with the enhanced electron density not only improves the capability for hydrogen activation, but also generates the partially negative-charged hydrogen species to suppress the electrophilic attack on the carbon-halogen bond and avoid the dehalogenation.

**Keywords:** Pd nanoparticles; hydrogenation; halonitrobenzenes; haloanilines

## 1. Introduction

Haloanilines are important intermediates of various fine and bulk chemicals in the chemical industry, such as medicines, pesticides, dyes and pigments [1,2]. Non-catalytic chemical reduction in halonitrobenzenes by iron powder [3] or alkali sulfide [4] can yield haloanilines with high selectivity. Unfortunately, these methods produce a large amount of residues, leading to serious environmental problems. Catalytic hydrogenation by supported metal catalysts is an environmentally friendly process for producing anilines, even in the presence of various other reducible functional groups (e.g., -C=C, -C=O) [5–8]. However, heterogeneous hydrogenation of halonitrobenzenes often suffers from serious dehalogenation due to the easy break of carbon-halogen (C-X) bonds on metal surfaces, subsequently lowering the yield of target products. Additionally, the dehalogenation produces a strong acidic environment in the reactor and thereby induces serious equipment corrosion. Thus, rationally designed metal catalysts combining high activity and selectivity are practically demanded to hydrogenate halonitrobenzenes into haloanilines.

As the typical hydrogenation catalysts, Pd nanocatalysts exhibit the excellent catalytic activity but poor selectivity for the hydrogenation of halonitrobenzenes due to easy dehalogenation. Many efforts have been developed to enhance their catalytic selectivity, including alloying [9], modulating the metal-support interaction [10,11] and tailoring sizes of metal catalysts [12–15]. However, these examples cannot well meet the requirements of large-scale hydrogenation of halonitrobenzenes to haloanilines. Although large Pd particles have been reported to improve the selectivity, the utilization efficiency of Pd is reduced [12]. Controlling the adsorption configuration of halonitrobenzenes on metal surfaces also can improve the hydrogenation selectivity. Recently, the fixation of Pd nanoparticles inside Beta zeolite crystals allowed the accessibility of the nitro group to Pd surface and exhibited the superior selectivity [16]. However, this method is still limited by the large internal

diffusion resistance of reactant molecules in the micropore of zeolite as well as the partially sacrificed Pd active sites. The precisely prepared single-atom Pd on MoC performed the selective hydrogenation of halonitrobenzenes [17]. However, it still lacks the cost-effective synthesis of catalysts for practical applications. Therefore, simply prepared heterogeneous catalysts are still challenging both in fundamental research and industrial need.

Besides those, the chemical states of the generated activated hydrogen species can be regulated by the electron density of metal nanoparticles [11,18]. Along this line, the partially negative-charged hydrogen species might tend to enable the selective nucleophilic attack on the nitro group rather than the electrophilic attack on the C-X bond. On this occasion, the dehalogenation could be maximally avoided, resulting in the enhanced selectivity of hydrogenation of halonitrobenzenes to haloanilines.

Herein, we demonstrate that the electronic structure of Pd nanoparticles effectively regulates the catalytic activity and selectivity for halonitrobenzenes-to-haloanilines hydrogenation. Layered double hydroxides (LDHs) were selected as typical basic supports, which could greatly enhance the electronic density of supported Pd nanoparticles.

## 2. Results and Discussion

Initially, the supports of CoFe-LDHs were synthesized by a hydrothermal method according to the previous report (Figure S1) [19]. As revealed from the X-ray diffraction (XRD) patterns (Figure S2), the diffraction peaks of the CoFe-LDHs are all well indexed to JPCDS No. 50-0235, indicating the successful formation of CoFe-LDHs. Then, the Pd nanoparticles were anchored on the surface of CoFe-LDHs by a photo-assisted deposition method. The Pd nanoparticles with an average size of 2.33 ± 0.2 nm (Figure S3a) are observed, as shown in the transmission electron microscope (TEM) image (Figure 1a). The measured lattice fringe spacing of 0.23 nm is assigned to the lattice spacing of the (111) plane of metallic Pd (Figure 1b). As the control catalyst, the Pd nanoparticles with average size of 3.06 ± 0.3 nm (Figure S3b) were also deposited on the surface of carbon black through the chemical coprecipitation method, which are confirmed by the TEM and high resolution transmission electron microscope (HRTEM) measurements (Figure 1c,d).

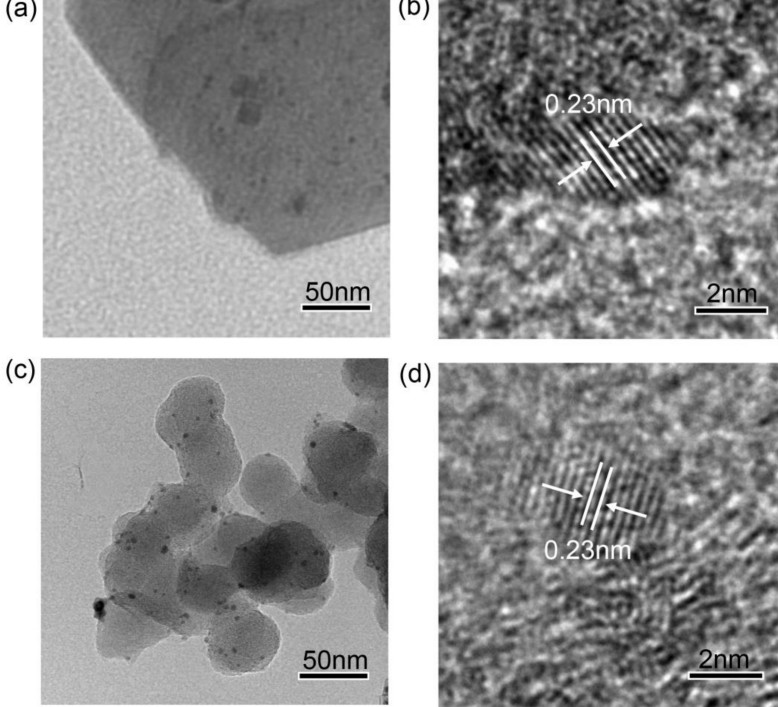

**Figure 1.** (**a**) TEM image of Pd/CoFe-LDHs catalyst; (**b**) HRTEM image of Pd/CoFe-LDHs catalyst; (**c**) TEM image of Pd/C catalyst; (**d**) HRTEM image of Pd/C catalyst.

Selective hydrogenation of *para*-chloronitrobenzene (*p*-CNB) to *para*-chloroaniline (*p*-CAN) was used as the model reaction to evaluate the catalytic performance of the present catalysts. After the optimization, the hydrogenation reactions were performed in isopropanol system at room temperature (25 °C) and 0.5 MPa of $H_2$ (Table S1). As shown in Figure 2a, a 97.4% conversion of *p*-CNB was achieved after 2 h hydrogenation by the Pd/CoFe-LDHs catalyst. While the Pd/C catalyst yielded the 98.7% conversion after 3 h under the optimized reaction conditions (Figure 2b). To further evaluate their intrinsic catalytic activity, the turnover frequency (TOF) values based on each exposed Pd atom were calculated by the Equations (S1) and (S2) [20]. The TOF value of Pd/CoFe-LDHs is 1706 $h^{-1}$, which is 1.8 times higher than that of Pd/C (962 $h^{-1}$, Figure 2c). Therefore, the Pd/CoFe-LDHs catalyst exhibited an enhanced catalytic activity, as compared with the Pd/C catalyst.

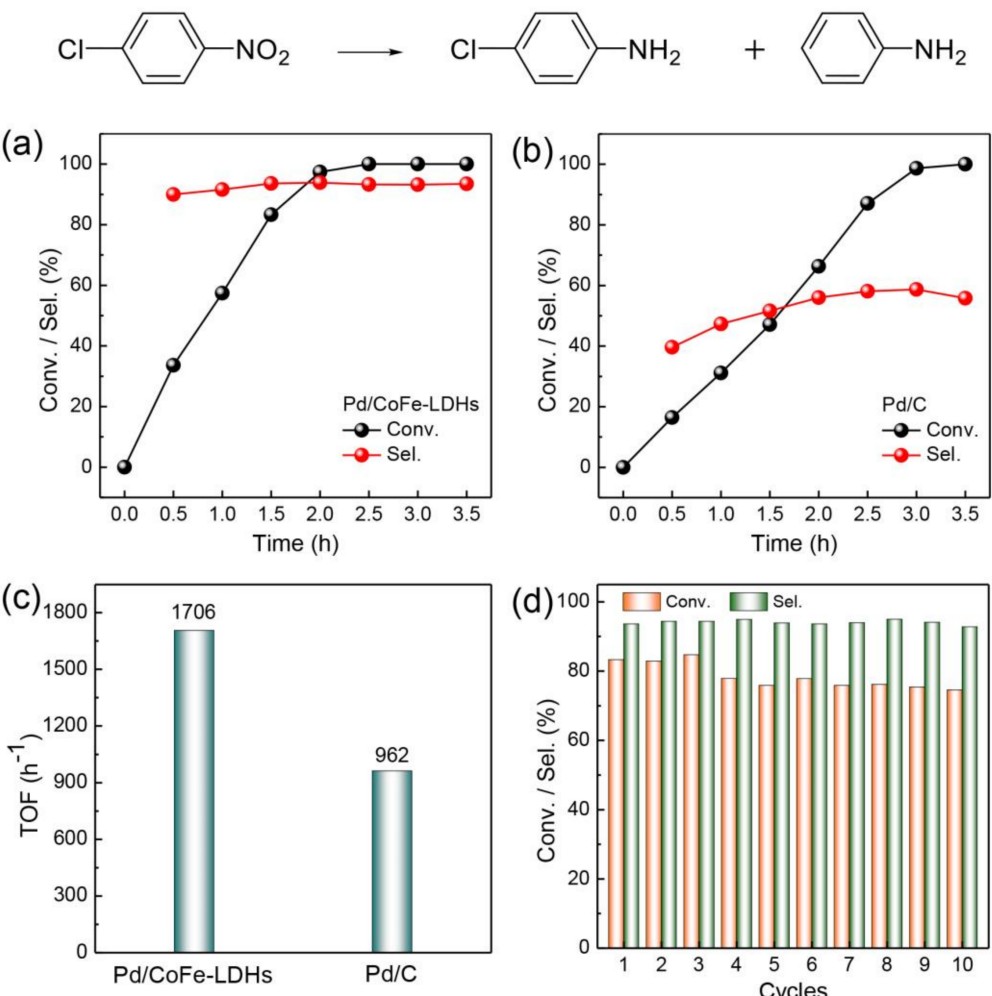

**Figure 2.** (**a**) Time course of conversion of *p*-CNB and selectivity of *p*-CAN catalyzed by Pd/CoFe-LDHs; (**b**) Time course of conversion of *p*-CNB and selectivity of *p*-CAN catalyzed by Pd/C; (**c**) TOF values based on each exposed Pd atom. (**d**) The cycling stability of Pd/CoFe-LDHs for the hydrogenation of *p*-CNB to *p*-CAN. Reaction conditions: catalysts (10.0 mg), *p*-CNB (1.0 mmol), isopropanol (2.0 mL), 25 °C, 0.5 MPa $H_2$.

Generally, the high catalytic activity is accompanied with the decreased selectivity due to the simultaneous hydrogenation of both the nitro group and C-X bond in previous reports [21,22]. Similarly, herein, the Pd/C catalyst shows a poor selectivity of 58.7% towards *p*-CAN, accompanied with dehalogenation products (Figure 2b). In contrast, the Pd/CoFe-LDHs catalyst exhibited the significantly improved selectivity to *p*-CAN, which reached

93.3% at near complete conversion of *p*-CNB (Figure 2a). More importantly, the selectivity of *p*-CAN for Pd/CoFe-LDHs was even preserved after another extended 1 h, indicating the greatly suppressed dechlorination. Comparatively, the further decreased selectivity and increased dechlorination for Pd/C were observed after another 0.5 h (Figure 2b). Therefore, the Pd/CoFe-LDHs catalyst exhibited both the enhanced catalytic activity and selectivity for hydrogenation of *p*-CNB to *p*-CAN, demonstrating its capability for anti-dechlorination.

Catalytic stability is another important factor to evaluate the applicability of the catalyst. After hydrogenation, the Pd/CoFe-LDHs catalyst was easily recycled by centrifugation and reused for the next cycle without any treatment. The *p*-CNB hydrogenation was performed at 1.5 h. As shown in Figure 2d, the Pd/CoFe-LDHs catalyst could keep its capacity of *p*-CNB conversion at the range from 74.6% to 84.7% for at least 10 cycles. Especially, the selectivity of *p*-CAN remained at around 94%. Therefore, the Pd/CoFe-LDHs catalyst with its preserved catalytic activity and selectivity demonstrate its excellent catalytic stability.

Understanding the different catalytic performance of the Pd/CoFe-LDHs and Pd/C catalysts is critical for the design and synthesis of novel heterogeneous catalysts for selective hydrogenation. Initially, the CoFe-LDHs supports had no catalytic activity for the hydrogenation of *p*-CNB before and after the same process as the preparation of Pd/CoFe-LDHs catalyst without adding Pd precursor. Thus, the possible influence of cations in CoFe-LDHs supports on the catalytic performance can be eliminated. Meanwhile, the size effect is also not the main factor for the improved catalytic selectivity of Pd/CoFe-LDHs. Previous literatures have reported that the selectivity of *p*-CAN increased with the increase in Pd nanoparticles sizes [12,23]. In contrast, herein, the Pd nanoparticles in CoFe-LDHs and Pd/C have similar sizes, and relative small Pd nanoparticles in Pd/CoFe-LDHs catalyst yielded the improved selectivity. Notably, the abundant surface hydroxyl groups on CoFe-LDHs supports are typical Lewis basic sites, which can greatly enhance the electronic density of the supported metal nanoparticles. Meanwhile, the carbon acts as the electron-withdrawing supports, on which electrons are transferred from metal nanoparticles to supports, leading to the decreased electronic density of metals [24]. Therefore, the different catalytic performance of Pd/CoFe-LDHs and Pd/C catalysts would be attributed to their different electronic structures.

To confirm this assumption, the electronic structures of Pd were examined by the X-ray photoelectron spectroscopy (XPS) analysis. The Pd 3d spectra of Pd/CoFe-LDHs and Pd/C catalysts are shown in Figure 3a. The binding energy of Pd $3d_{5/2}$ for Pd/CoFe-LDHs is 335.45 eV, which is lower than that of 335.80 eV for Pd/C, the fraction of surface $Pd^0$ on Pd/CoFe-LDHs (80.1%) is also higher than that on Pd/C (67.9%). Therefore, Pd nanoparticles supported on CoFe-LDHs with the rich surface hydroxyl groups exhibit the higher electronic density.

Previous reports have proved that the Pd nanoparticles with enhanced electronic density exhibited high ability for hydrogen dissociation, further resulting in the improved catalytic hydrogenation activity [25,26]. Then, the kinetic experiments were also performed to further explore their intrinsic catalytic ability for hydrogenation. As shown in Figures S4 and S5, the concentration of *p*-CNB decreased linearly with time at each temperature, which is in accordance with the characteristics of zero order reaction. Thus, the selectively catalytic hydrogenation of *p*-CNB is not limited by the external diffusion of reactants. The apparent activation energies (Ea) of the two catalysts can be derived by Arrhenius equation (Equation (1)):

$$\ln k = -Ea/RT + \ln A \qquad (1)$$

where k is reaction rate constant, A is the pre-exponential factor, R is the molar gas constant, and T is the thermodynamic temperature. As shown in Figure 3c, we determined Ea to be 27.2 kJ mol$^{-1}$ for the Pd/CoFe-LDHs catalyst and 35.0 kJ mol$^{-1}$ for the Pd/C catalyst. The lower Ea of the Pd/CoFe-LDHs catalyst suggests its higher intrinsic activity for the catalytic hydrogenation of *p*-CNB.

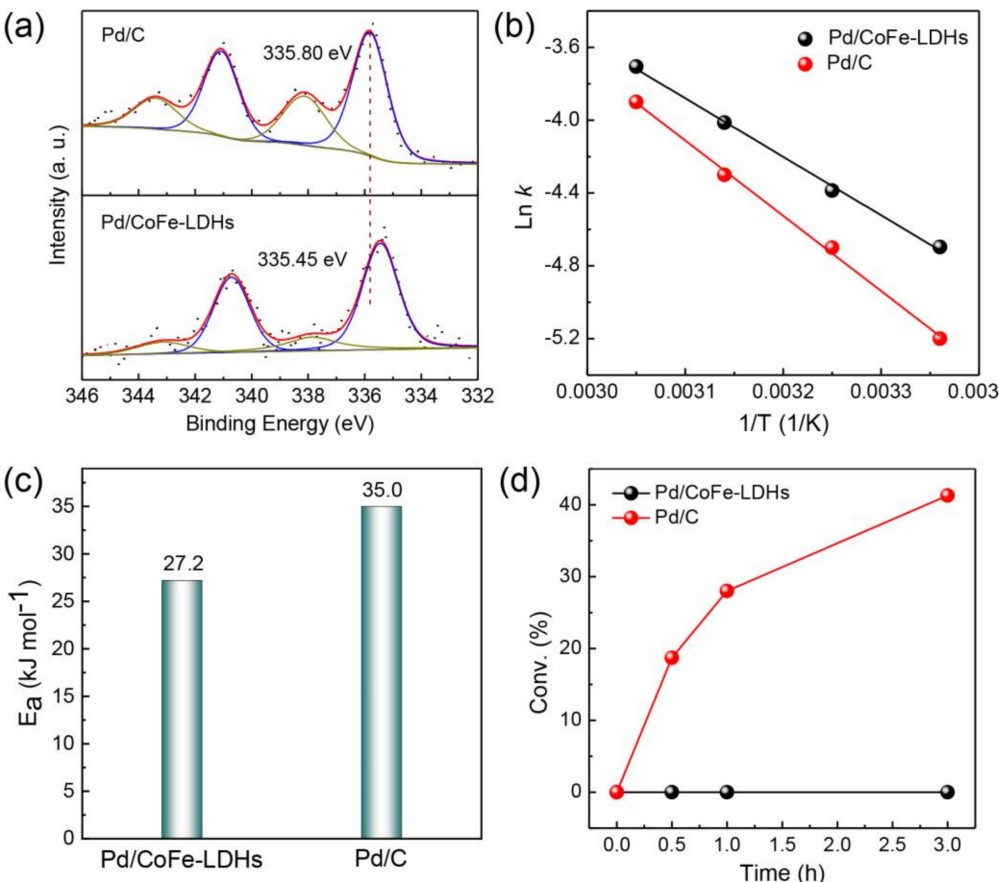

**Figure 3.** (**a**) XPS analysis of Pd/C and Pd/CoFe-LDHs. (**b**) Plot of lnk vs. 1/T for Pd/CoFe-LDHs and Pd/C, derived from *p*-CNB reaction rates vs. time. (**c**) Ea of Pd/CoFe-LDHs and Pd/C. (**d**) Catalytic hydrogenation of chlorobenzene by Pd/CoFe-LDHs and Pd/C. Reaction conditions: catalysts (10.0 mg), chlorobenzene (1.0 mmol), isopropanol (2.0 mL), 25 °C, 0.5 MPa $H_2$.

Previous studies have indicated that the hydrogenolysis of C-X bond is attributed to the electrophilic attack by the cleaved hydrogen on the C-X bond, while the reduction in the nitro group was a nucleophilic attack process [10,18,27]. Generally, $H_2$ can be easily dissociated under a mild condition by Pd nanoparticles, and the generated two activated hydrogen (H) can attack various functional groups through both nucleophilic and electrophilic processes. As the metal nanoparticles with high electronic density, the activated hydrogen is transformed into partially negative-charged hydrogen species, owing to the donated d-electrons of metal to the σ* antibonding orbital of $H_2$ [25,28]. Differently from the H, the partially negative-charged hydrogen species tend to undergo a nucleophilic attack on the nitro group rather than the electrophilic attack on the C-X bond. To experimentally prove this assumption, both the Pd/C and Pd/CoFe-LDHs catalysts were used to catalyze the hydrogenation of chlorobenzene under the same reaction conditions. As shown in Figure 3d, the Pd/CoFe-LDHs catalyst with higher activity for hydrogenation of *p*-CNB exhibited the barely catalytic activity for chlorobenzene transformation at 25 °C. In contrast, the Pd/C catalyst successfully hydrogenated chlorobenzene along with 100% selectivity of hydrogenolysis, indicating its strong activity for dechlorination.

According to the above analysis, a possible catalytic process for *p*-CNB hydrogenation by Pd/CoFe-LDHs could be proposed as shown in Figure 4. The high electronic density of Pd nanoparticles on CoFe-LDHs can significantly promote the hydrogen activation, generating the partially negative-charged active hydrogen species. Then, the nitro group in *p*-CNB molecule could preferentially undergo a nucleophilic attack by the partially

negative-charged active hydrogen species through hydrogen spillover on the surface of Pd nanoparticles.

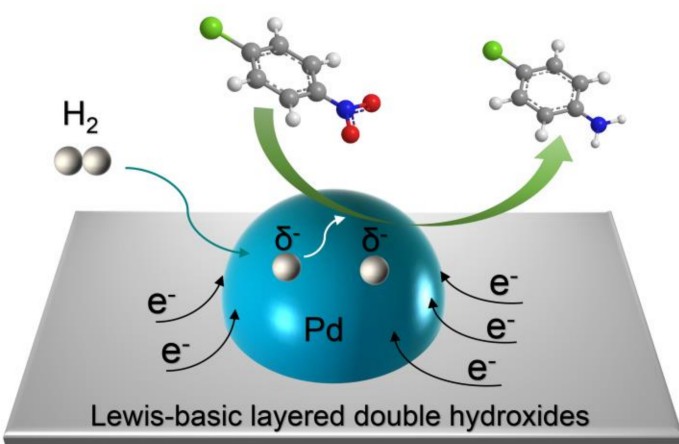

**Figure 4.** Schematic diagram of *p*-CNB hydrogenation catalyzed by Pd/CoFe-LDHs catalyst.

The scope of the substituted nitroarenes were further investigated by the Pd/CoFe-LDHs catalyst under 0.5 MPa H$_2$ pressure at low temperatures (Table 1, GC-MS spectra are showed in Figure S6). Similar to *p*-CNB, *ortho*-chloronitrobenzene was nearly completely hydrogenated within 2 h with a selectivity of 91.5% (Entry 1). For the catalytic hydrogenation of *meta*-chloronitrobenzene, the conversion and selectivity were 98.7% and 92.8% after 4 h reaction, respectively. Meanwhile, 3-fluoro-nitrobenzene and 4-fluoro-nitrobenzene were converted completely at 2 h and 2.5 h with 99% selectively toward their target products, respectively (Entries 3 and 4). Unfortunately, the Pd/CoFe-LDHs catalyst cannot achieve the selective hydrogenation of aromatic nitro compounds with bromide/iodide as substituent group, owing to their weak bond energy of C-Br and C-I bonds. For example, only 55.6% selectivity of 4-bromoaniline was yielded when the conversion of 4-bromonitrobenzene was 97.7% under the same reaction conditions. As expected, for the nitrobenzene with other substituent groups, such as -OCH$_3$, -CN, -NH$_2$, -OH and -COOH, the Pd/CoFe-LDHs catalyst also exhibited excellent selectivity of the corresponding anilines with high catalytic activity (Entries 5–10). It should be noted that due to the solubility of the substrates or products, the amount of substrates for these reactions was halved and the reaction temperature was raised to 60 °C. Therefore, the Pd/CoFe-LDHs catalyst demonstrates a satisfactory scope in possible practical application.

**Table 1.** Hydrogenation of various substituted nitroaromatics.

| Entry | Substrates | Products | Time (h) | Conv. (%) | Sel. (%) |
|---|---|---|---|---|---|
| 1 | Cl—C$_6$H$_4$—NO$_2$ (ortho) | Cl—C$_6$H$_4$—NH$_2$ (ortho) | 2 | >99.9 | 91.5 |
| 2 | Cl—C$_6$H$_4$—NO$_2$ (meta) | Cl—C$_6$H$_4$—NH$_2$ (meta) | 4 | 98.7 | 92.8 |
| 3 | F—C$_6$H$_4$—NO$_2$ (meta) | F—C$_6$H$_4$—NH$_2$ (meta) | 2 | >99.9 | 99 |
| 4 | F—C$_6$H$_4$—NO$_2$ (para) | F—C$_6$H$_4$—NH$_2$ (para) | 2.5 | >99.9 | 99 |
| 5 [a] | CH$_3$O—C$_6$H$_4$—NO$_2$ | CH$_3$O—C$_6$H$_4$—NH$_2$ | 1 | >99.9 | 99 |

**Table 1.** *Cont.*

| Entry | Substrates | Products | Time (h) | Conv. (%) | Sel. (%) |
|---|---|---|---|---|---|
| 6 [a] | | | 1 | >99.9 | 99 |
| 7 [a] | | | 1 | >99.9 | 99 |
| 8 [a] | | | 1 | >99.9 | 99 |
| 9 [a] | | | 1.5 | >99.9 | 99 |
| 10 [a] | | | 3 | 89.2 | 99 |

Reaction conditions: catalysts (10.0 mg), substrates (1.0 mmol), isopropanol (2.0 mL), 25 °C, 0.5 MPa $H_2$.
[a] Substrates (0.5 mmol), 60 °C.

## 3. Materials and Methods

### 3.1. Materials

All of the chemical reagents for the synthesis and reactions were commercially available and used without further purification. Cobalt nitrate hexahydrate (Energy Chemical, 99%, Shanghai, China), Ferric nitrate nine hydrate (Energy Chemical, 99%), sodium hydroxide (Alfa Aesar, 98%, Shanghai, China), sodium carbonate (Macklin, 98%, Shanghai, China), *para*-chloronitrobenzene (Energy Chemical, 98%), *ortho*-chloronitrobenzene (Rhawn, 99%, Shanghai, China), *meta*-chloronitrobenzene (Aladdin, 98%, Shanghai, China), 3-fluoro-nitrobenzene (Energy Chemical, 98%), 4-fluoro-nitrobenzene (Energy Chemical, 99%), 4-nitroanisole (Ourchem, 98%, Sinopharm Chemical ReagentCo., Ltd, Shanghai, China), 4-nitrobenzonitrile (Energy Chemical, 98%), 4-nitroaniline (Energy Chemical, 99%), 2-nitrophenol (Energy Chemical, 98%), 4-nitrobenzoic acid (Energy Chemical, 98%), 3-nitrophenol (Energy Chemical, 98%).

### 3.2. Catalyst Preparation

Synthesis of CoFe-LDHs supports: CoFe-LDHs supports were synthesized by a simple hydro-thermal method [19]. Typically, $Co(NO_3)_2 \cdot 6H_2O$ and $Fe(NO_3)_3 \cdot 9H_2O$ (the molar ratios of Co/Fe was 2:1) were dissolved in the same beaker with deionized water, and the concentrations were 0.6 M and 0.3 M, respectively. Similarly, NaOH (0.192 M) and $Na_2CO_3$ (0.8 M) were also dissolved in the same beaker with deionized water, respectively. Then, two solutions of equal volume (30 mL) were poured into a beaker under vigorous stirring. Afterwards, the mixed solution was transferred into a stainless steel Teflon-lined autoclave (100 mL) at 80 °C for 48 h. After natural cooling to room temperature, the precipitation was centrifuged off, alternatively washed with water and ethanol three times, and then dried at 60 °C for the future use.

Synthesis of the Pd/CoFe-LDHs catalyst: The Pd/CoFe-LDHs catalyst was prepared by a photo-assisted deposition method. Briefly, 300 mg of CoFe-LDHs supports were suspended in a mixture of deionized water (30 mL) and methanol (10 mL). After adding 7.58 mg of $Pd(NO_3)_2$ (the Pd mass fraction was 39.6%), the solution was degassed with Ar for 30 min and then irradiated for 3 h under a Xe lamp. The obtained catalyst was centrifuged off, washed with water and dried overnight in a vacuum oven.

Synthesis of the Pd/C catalyst: The Pd/C catalyst was prepared by chemical coprecipitation method. Briefly, 300 mg of carbon black were suspended in 30 mL deionized water, and then 7.58 mg of $Pd(NO_3)_2$ (the Pd mass fraction was 39.6%) was added with stirring for 30 min. After adding 300 mg of urea, the mixture solution was heated to 60 °C and kept under stirring for 1.5 h to precipitate Pd ions. When the solution was naturally cooled to

room temperature, 30 mg of $NaBH_4$ was added to reduce Pd. Finally, the obtained catalysts were centrifuged off, washed with water and dried overnight in a vacuum oven.

### 3.3. Catalyst Characterization

X-ray diffraction (XRD) measurements of the catalysts were performed using a Shimadzu (Kyoto, Japan) X-ray diffractometer (Model 6000) equipped with a Cu K$\alpha$ radiation source. The microstructures of the catalysts were studied by transmission electron microscope (TEM, Hitachi HT-7700, Tokyo, Japan) with an accelerating voltage of 120 kV. High resolution transmission electron microscopy (HRTEM) images were acquired on a JEOL JEM-F200 microscope (Tokyo, Japan) with an accelerating voltage of 200 kV. X-ray photoelectron spectroscopy (XPS) profiles were obtained on a Thermo Fisher ESCALAB Xi+ (Waltham, MA, USA) equipped with an Al K$\alpha$ X-ray source. The actual contents of Pd in the catalysts were determined by inductive coupled plasma spectrometry (ICP, NexION 350D, PerkinElmer, Waltham, MA, USA).

### 3.4. Catalytic Hydrogenation

Liquid-phase selective hydrogenation of *p*-CNB was carried out in a stainless steel autoclave equipped with the pressure and temperature control system. For a typical catalytic reaction, 1.0 mmol of *p*-CNB and 10.0 mg of catalyst were mixed in 2.0 mL of isopropanol. The reactor was purged with pure $H_2$ (0.5 MPa) three times to replace air in the system and then processed with the desired $H_2$ pressure (0.5 MPa) and stirring rate (700 r min$^{-1}$) at room temperature (25 °C). The products were analyzed by gas chromatography (GC, Agilent 7890B, Santa Clara, CA, USA) using an instrument equipped with a flame ionization detector (FID) and a mass spectrometer (MS).

## 4. Conclusions

The catalytic performance of supported metal nanoparticles for hydrogenation has been successfully demonstrated to be determined by their electronic structures. For the selective hydrogenation of *p*-CNB to *p*-CAN, the Pd/CoFe-LDHs catalyst exhibited the simultaneously enhanced catalytic activity and selectivity, achieving a 93.3% selectivity of *p*-CAN at the near complete conversion of *p*-CNB with TOF of 1706 h$^{-1}$ under mild conditions of 0.5 MPa $H_2$ and 25 °C. Mechanism studies reveal that Pd nanoparticles with the high electronic density facilitate the hydrogen dissociation and generate the partially negative-charged hydrogen species, suppressing the electrophilic attack on the C-X bond and thereby exhibiting the simultaneously enhanced catalytic activity and selectivity for the selective hydrogenation of halonitrobenzenes to haloanilines.

**Supplementary Materials:** The following are available online at https://www.mdpi.com/article/10.3390/catal11050543/s1, Figure S1: TEM image of CoFe-LDHs supports, Figure S2: XRD patterns of the CoFe-LDHs supports and Pd/CoFe-LDHs catalyst, Figure S3: Size distribution of Pd nanoparticles in Pd/CoFe-LDHs and Pd/C catalysts, Figure S4: Curves of p-CNB conversion at different temperatures, Figure S5: Curves of p-CNB content with reaction time, Figure S6: The GC-MS spectra of hydrogenation of various substituted nitroaromatics, Table S1: Optimization of reaction conditions, Equation (S1): The equation of Pd dispersion, Equation (S2): The equation of TOF values.

**Author Contributions:** Conceptualization, Y.Q. and S.Z.; experiments and writing-original draft, Z.L.; writing—review and editing, Y.Q., S.Z. and M.Z.; funding acquisition, Y.Q. and S.Z. All authors have read and agreed to the published version of the manuscript.

**Funding:** This research was funded by National Natural Science Foundation of China (21872109 and 22002115), Youth Talent Support Project from China Association of Science and Technology, and Natural Science Basic Research Plan in Shaanxi Province of China (2019JQ-039).

**Data Availability Statement:** All relevant data are contained in the present manuscript. Other inherent data are available on request from the corresponding author.

**Acknowledgments:** The characterization of the catalysts is supported by Frontier Institute of Science and Technology and Instrumental Analysis Center of Xi'an Jiaotong University.

**Conflicts of Interest:** The authors declare no conflict of interest.

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
