# Peer review of "Electron-Enriched Pd Nanoparticles for Selective Hydrogenation of Halonitrobenzenes to Haloanilines"

_catalysts, doi:10.3390/catal11050543_

Round 1

Reviewer 1 Report

The manuscript from Dr Qu and co-workers reported on Pd nanoparticles selective hydrogenation of halonitrobenzenes to haloanilines. I should note that this manuscript described and summarized fully support the authors’ assertions.  Finally, I recommend acceptance of the manuscript with minor correction. Some comments are listed

  1. Line 50: typo error for ‘halonitrobenzenes’
  2. Line 182: it needs to mention ‘hydrogen spillover’.
  3. Figure 1: suggests not to insert the size distribution of PD nanoparticles in the figures (put in supporting information)
  4. Figure 4: suggests to add a negative sign to ‘e-
  5. Figure S2: missing of X-axis label

Author Response

We thank the reviewer' constructive comments on our manuscript, which can improve the quality of our manuscript.

The response of each comment is listed in the attachment.

Reviewer 2 Report

This manuscript deals with a topic which is certainly interesting and suitable for Catalysts journal. It is clearly written and structured. The quality of figures and Tables is good and the bibliographic references adequate. The work represents a nice contribution in the field of synthetic chemistry. The manuscript can be accepted for publication but the authors should first make some corrections as follows:

1) The last sentence of the introduction (lines 63-67) is more appropriate for the conclusions section as it anticipates specific results obtained in this work. I recommend moving it to the Conclusions, also because this section by the moment is giving just qualitative information, so concrete data to illustrate the main conclusions would be appreciated.

2) Although English in general is good there are several deficiencies that should be corrected:

  • line 17: improves
  • line 18: generates
  • line 24: "Haloanilines are important intermediates..."
  • Cancel the article the in lines 28, 31, 35, 51 (before "simply"), 153 (before "high") and 156 (before "Figure").
  • Insert the article the in lines 46 (before "nitro"), 57 (before "nitro" and "C-X bond"), 109 (before "catalyst"), 168 (before "nitro") and 251 (before "catalysts").
  • line 44: is reduced
  • line 50: performed the selective hydrogenation
  • line 90: exhibited an enhanced catalytic activity
  • line 107: demonstrating its capability
  • line 112: "the Pd/CoFe-LDH catalyst could keep its capacity of p-CNB conversion"
  • line 156: Figures
  • lines 170 and 176: cancel of after both
  • line 197: "toward their target products"
  • line 198: insert of after hydrogenation
  • line 227: "the Co/Fe molar ratio was 2:1"
  • lines 238: and 243 "the Pd mas fraction was 39.6%"
  • line 251: were studied by transmission
  • lines 252-253: "High Resolution Transmission Electron MIcroscopy (HRTEM) images were acquired on a JEOL..."
  • line 264: at room temperature
  • line 265: "The products were analyzed by gas chromatography (GC) using an instrument equipped with flame ionization detector (FID) and a mass spectrometer (MS)"
  • line 268: Cancel In summary,

Other typos:

  • line 287: Technology
  • line 290: nitroaromatic

In addition, the authors should provide the trademark and model of the gas chromatograph employed.

Author Response

We thank the reviewer' constructive comments on our manuscript, and apologize for the spelling errors in the manuscript. 

The response of each comment is listed in the attachment.

Reviewer 3 Report

The manuscript catalysts-1190518 describes the synthesis of electron-enriched Pd nanoparticles for selective hydrogenation of halonitrobenzenes to haloanilines.

The manuscript is well-written; some minor comments are listed below.

  1. Section 3.2, first paragraph: Were Co(NO3)2·6H2O and Fe(NO3)3·9H2O dissolved together (in the same beaker)? Please, clarify.
    2. Section 3.2, first paragraph: The same for NaOH and Na2CO3.
    3. Section 3.2, first paragraph: References for this synthesis are necessary.
    4. Section 3.2, second paragraph: Please, provide references for the synthesis of Pd/CoFe-LDHs catalyst.
    5. Section 3.2, third paragraph: An explanation of the addition of urea is needed.
    6. Section 3.2, third paragraph: An explanation of the addition of NaBH4 is needed.
    7. Section 3.2, third paragraph: References are needed regarding the synthesis of Pd/C catalyst.
    8. Page 3, last paragraph: What happens to the catalyst after the ten cycles (chemical/structural modifications) and fails to work? An explanation is needed here.

Author Response

(The authors gave the same response as above.)
